# Thin-Film Transistor Digital Microfluidics Circuit Design with Capacitance-Based Droplet Sensing

**DOI:** 10.3390/s24154789

**Published:** 2024-07-24

**Authors:** Shengzhe Jiang, Chang Li, Jiping Du, Dongping Wang, Hanbin Ma, Jun Yu, Arokia Nathan

**Affiliations:** 1School of Information Science and Engineering, Shandong University, Qingdao 266237, China; 202312640@mail.sdu.edu.cn (S.J.); 202100120110@mail.sdu.edu.cn (C.L.); 202100120037@mail.sdu.edu.cn (J.D.); arokia.nathan@sdu.edu.cn (A.N.); 2CAS Key Laboratory of Bio-Medical Diagnostics, Suzhou Institute of Biomedical Engineering and Technology, Chinese Academy of Sciences, Suzhou 215163, China; hanbin.ma@acxel.com; 3Guangdong ACXEL Micro & Nano Tech Co., Ltd., Foshan 528299, China; 4Darwin College, University of Cambridge, Cambridge CB3 9EU, UK

**Keywords:** active matrix, digital microfluidics, thin-film transistor, capacitance sensing, droplet position sensing

## Abstract

With the continuous expansion of pixel arrays in digital microfluidics (DMF) chips, precise droplet control has emerged as a critical issue requiring detailed consideration. This paper proposes a novel capacitance-based droplet sensing system for thin-film transistor DMF. The proposed circuit features a distinctive inner and outer dual-pixel electrode structure, integrating droplet driving and sensing functionalities. Discharge occurs exclusively at the inner electrode during droplet sensing, effectively addressing droplet perturbation in existing sensing circuits. The circuit employs a novel fan-shaped structure of thin-film transistors. Simulation results show that it can provide a 48 V pixel voltage and demonstrate a sensing voltage difference of over 10 V between deionized water and silicone oil, illustrating its proficiency in droplet driving and accurate sensing. The stability of threshold voltage drift and temperature was also verified for the circuit. The design is tailored for integration into active matrix electrowetting-on-dielectric (AM-EWOD) chips, offering a novel approach to achieve precise closed-loop control of droplets.

## 1. Introduction

Digital microfluidics (DMF) based on the electrowetting-on-dielectric (EWOD) principle, a promising subfield of microfluidics, has emerged as a pivotal technique for advancing field-programmable liquid arrays in a two-dimensional plane [1,2,3]. The wettability of the solid–liquid interface can be manipulated according to the electrode voltage in EWOD DMF. This modulation in wettability, driven by differences in surface tension, facilitates the movement of droplets between electrodes. By applying sequential voltages to the electrode array via the peripheral circuitry, precise control of droplets can be achieved along pre-programmed routes [4]. Conventional DMF chips employ a passive matrix architecture, posing substantial challenges for scalability attributed to constraints in wiring space.

Thin-film transistor (TFT)-based active matrix (AM) technology effectively resolves this issue, providing an optimal solution for implementing large-scale field-programmable droplet arrays [5,6,7]. Much like flat-panel display technology, AM-EWOD utilizes a row–column scanning approach to deliver precise driving voltages, thereby minimizing the required signal lines. Given its exceptional configurability, integrity, and operational efficiency, AM-EWOD is promising for applications in antibody technology, proteomics, and single-cell analysis [8,9,10].

Surface non-uniformities, such as dust particles, scratches, and residual reagents, impede the movement of droplets between electrodes in AM-EWOD [11]. Open-loop control systems lack the feedback information necessary to correct droplet perturbations, thereby compromising system performance. With the rapid advancement in microanalyses and detection, droplet samples exhibit higher throughput, imposing increasingly stringent demands on the precise control of droplets. Developing a feedback system to achieve real-time, fast, and precise droplet position sensing is crucial for achieving a stable AM-EWOD system [12]. The mainstream droplet sensing methods primarily include optical sensing and electrical sensing [13,14]. Due to the high cost, bulky size, and difficulties in capturing transparent and light-sensitive liquid samples, optical sensing methods face limitations in their application in AM-EWOD systems [13]. Electrical sensing possesses distinctive advantages, including high sensitivity, rapid response speed, and high integrity, primarily encompassing resistive and capacitive sensing [15,16]. Resistive sensing involves direct physical contact between droplets and microelectrodes, which can potentially lead to the contamination of the electrode surface and thereby affect the accuracy of the detection results [17]. Capacitive sensing utilizes driving electrodes beneath the droplets, ensuring complete compatibility with AM-EWOD while preserving droplet integrity without contamination or morphology alteration [18]. 

This paper proposes a novel capacitive-based droplet position sensing circuit, and the output performance and stability are also investigated. This design is fully compatible with TFT-based AM-EWOD and can provide both driving signals for droplets and real-time position feedback for the system. The circuit employs a unique dual-pixel electrode structure, where discharge occurs exclusively on the inner electrode during droplet position sensing, effectively addressing the disturbance issues encountered in existing sensing circuits. The circuit incorporates a novel fan-shaped structure of TFTs designed for higher voltage stability and enhanced performance. This circuit can supply a pixel voltage of 48 V and exhibits a sensing voltage difference exceeding 10 V between deionized water and silicone oil, highlighting its exceptional droplet driving and precise sensing capabilities. The circuit further validates threshold voltage drift and temperature stabilities. This design introduces a novel approach to enable arrayed droplet position sensing and precise closed-loop control in AM-EWOD. 

## 2. Circuit Design

Figure 1a illustrates the structure of an AM-EWOD device. The enclosed AM-EWOD device features a sandwich structure with a top plate and a substrate made from glass, separated by a spacer defining the gap distance. The TFT layer is fabricated on the glass substrate using standard amorphous silicon hydrogenated (a-Si:H) TFT technology, commonly found in flat-panel displays [19]. Gate-on-array (GOA) circuits, pixel circuits, and the proposed droplet sensing circuit are all manufactured within this layer, facilitating seamless integration with AM-EWOD [20,21]. Pixel electrodes are connected to the TFT layer, supplying driving voltage to droplets. The 300 nm SiN_x_ dielectric layer manipulates interfaces and prevents droplet electrolysis. The dielectric layer and common electrode were spin-coated with a 200 nm hydrophobic layer with Teflon AF1600 2% (Amorphous Fluoroplastics) from Chemours Company to enhance droplet mobility. Droplets are sandwiched between two hydrophobic layers, thus forming an equivalent capacitor. Pixel electrodes apply driving signals, while the common electrode is grounded. The significant difference in the dielectric constants between water (liquid sample) and silicone oil (surrounding medium) results in a considerable difference in equivalent capacitances, enabling capacitive-based droplet sensing. 

A simplified circuit model for capacitive-based droplet position sensing is depicted in Figure 1b. In Figure 1a, pixel electrodes function as the upper plate and are connected to the TFT layer, while common electrodes act as the lower plate and are grounded, with the liquid sample serving as the dielectric medium. These elements form the equivalent capacitor depicted in Figure 1b. The remaining components in Figure 1b, including the sensing capacitor, switches, and signal lines, are all integrated within the TFT layer, providing sensing and driving signals to the pixel electrode under the influence of external driving signals. In the droplet driving mode, S_1_ is closed, while S_2_, S_3_, and S_4_ are open, allowing driving signals to be applied to pixel electrodes. Individual droplets can be independently driven through sequential voltage applied by row–column scanning signals. On-chip GOA circuits provide serial row scanning signals, while parallel column scanning signals are supplied by peripheral circuits, transmitted to the chip via flexible printed circuit boards (FPC). In the droplet sensing mode, S_1_ and S_4_ are open, while S_2_ and S_3_ are closed. Under the influence of an external signal RWS, the sensing capacitor charges, leading to an increase in voltage on the lower plate. Due to the equivalent capacitance variance of samples and the surrounding medium, charging results in varying output voltages, making it feasible to sense the presence of droplets in their current positions [22]. 

There is an issue of pixel capacitor discharge during the sensing processor existing passive capacitance-based droplet sensing circuits [23]. Moreover, the requirement to reset the sensing capacitor before each detection in conventional active methods also leads to the discharge of the pixel capacitor. During the discharge of pixel electrodes, a sudden reduction in the electric field can alter the field distribution on the droplet, potentially causing changes in the droplet’s shape, such as contraction or expansion. The abrupt decrease in voltage may lead to instability in the droplet within the electric field, resulting in irregular vibrations or jitter. This instability could make it challenging for the droplet to remain in its intended position, potentially causing it to move freely to a new location on the chip and affecting the experimental process [16]. Once the electric field support is lost, positioning the droplet may become difficult, particularly in applications requiring high-precision droplet manipulation, such as sample separation, mixing, and reactions. The novel dual-electrode pixel unit structure proposed, as shown in Figure 2a, addresses this issue. Each pixel electrode measures 100 × 100 μm^2^ and is divided into two regions: the inner sensing region (with an equivalent capacitance of C_po_) and the outer driving region (with an equivalent capacitance of C_pi_). 

The corresponding droplet position sensing unit is depicted in Figure 2b, comprising eight TFTs and two capacitors. C_sense_ in Figure 2b corresponds to the sensing capacitor shown in Figure 1b and it is integrated within the TFT layer. The only difference for the droplet sample compared to the surrounding medium is the equivalent capacitance. The proposed circuit is divided into a driving module and a detecting module, separated by T_3_. The driving module consists of a 2T1C pixel circuit. In the driving mode, with T_3_ on, the driving voltage is applied to C_po_ and C_pi_. This configuration ensures that the electrode’s inner and outer regions receive pixel voltages, facilitating stable droplet manipulation. In the detection mode, T_3_ is disconnected, isolating C_po_ from C_pi_. Before the commencement of detection, the RST signal initiates, resulting in the discharge of the pi node to a low level, while the po node remains high due to the isolation provided by T_3_. At this stage, concerning the pixel electrode, the external region maintains a stable state with a sustained high voltage, effectively suppressing internal electrode discharge-induced disturbances. This stability ensures that the droplet remains securely positioned on the electrode. Under the influence of the external signal RWS, the voltage at the pi node increases, as described by the following equation: (1)$$V_{pixel}=V_{RWS}\frac{C_{sense}}{C_{sense}+C_{pi}}$$

The equivalent capacitance of deionized water and silicone oil can be derived using the flat-plate capacitor formula: (2)$$C_{DI\;water}=\frac{\varepsilon_{water}\varepsilon_{0}S}{d}\approx \frac{78\times 8.854\times 10^{-12}\times 5\times 10^{-9}}{3\times 10^{-5}}\approx 0.12\;pF$$
(3)$$C_{silicone\;oil}=\frac{\varepsilon_{oil}\varepsilon_{0}S}{d}\approx \frac{1\times 8.854\times 10^{-12}\times 5\times 10^{-9}}{3\times 10^{-5}}\approx 0.0015\;pF$$

The sensing mechanism of the proposed circuit primarily involves droplet position sensing. The difference in equivalent capacitance between the liquid sample (e.g., DI water) and the surrounding medium (e.g., silicone oil) results in varying sensor outputs, allowing for the distinction between the two and enabling the droplet sample’s position sensing on the chip. This position information allows for the calibration of droplet placement and enables stable closed-loop control. A 0.1 pF was chosen as C_sense_ to amplify the discernibility of the output discrepancies. T_4_ and T_5_ are controlled by the serial row signal Scan and parallel column signal Ctrl, respectively, enabling arrayed sensing. When the array is configured as M (rows) × N (columns), only M Gate, Scan, and RWS signals, along with N Ctrl and Data signals, are required to drive and sense droplets in the array. 

To achieve stable droplet manipulation in AM-EWOD, effective driving voltages reaching tens or even hundreds of volts are necessary, imposing stringent demands on the high-voltage performance of TFTs [24]. In widely commercialized TFT processes, a-Si operates at significantly higher voltages than low-temperature polysilicon (LTPS) and Indium Gallium Zinc oxide (IGZO) [25,26]. It has been confirmed to possess sufficient high-voltage stability, meeting the requirements of AM-EWOD applications. Furthermore, due to its cost-effectiveness, straightforward manufacturing process, and excellent uniformity, a-Si has found extensive application in AM-EWOD. However, in conventional bottom structure a-Si TFTs, high drain current stress under large gate voltages can induce self-heating effects, thereby deteriorating device characteristics [27]. Drain-offset TFT alleviates this issue; however, it concurrently increases the on-state resistance [28]. 

To further enhance high-voltage performance, a novel fan-shaped TFT structure was adopted in this study, as depicted in Figure 3a. This structure incorporates a floating electrode between the source and the drain. Compared to conventional single-channel TFTs, the floating electrode in the fan-shaped structure can form an additional electric field control layer between the source and drain, which helps establish a balanced electric field within the channel. This design reduces the lateral electric field intensity, thereby minimizing the hot carrier effect and migration barriers caused by uneven electric fields, ultimately enhancing the stability of the TFT. Figure 3b,c illustrate the typical transfer and output characteristics of the a-Si:H TFTs (W/L = 10 μm/6 μm) used in this work. Figure 3d–f compare the positive bias stability between conventional TFTs and fan-shaped TFTs. Under the same bias conditions (V_g_ = 50 V, V_d_ = 40 V), the fan-shaped TFT exhibits a smaller V_th_ shift, with ΔV_th_ remaining less than 0.2 V over 3000 s. This is because the uniform electric field reduces the formation of trap states in amorphous silicon, thereby minimizing the V_th_ shift caused by these trap states. Additionally, the uniform electric field helps stabilize charge migration paths, reducing the V_th_ shift resulting from uneven charge distribution. 

## 3. Results and Discussion

A COMSOL simulation of the chip cross-section was conducted, to validate the stability of the dual-electrode pixel structure, as depicted in Figure 4. Three pixel electrodes were configured with a pixel pitch of 5 μm. Several factors must be considered when selecting the size of the inner electrode. If the electrode area is too small, the equivalent capacitance of the inner electrode will decrease, resulting in lower sensing accuracy and making detection more challenging. Conversely, if the electrode area is too large, the outer electrode becomes too narrow, causing the droplet to experience excessive internal pressure during detection, which compromises system stability. Therefore, the inner electrode structure depicted in Figure 3a was chosen. In the simulation, an inner electrode length of 45 μm was employed to increase the electrode area while ensuring droplet stability. 

Figure 4a illustrates the change in the droplet volume fraction, while Figure 4b depicts the variation in system potential. In Figure 4a, small electrodes on either side within each pixel symbolize outer ring electrodes, each with a width of 27.5 μm. The central electrode denotes the inner sensing electrode, positioned at 1 μm from the outer electrodes. At t = 0 s, corresponding to the initial state, no voltage is applied to the electrodes, and the droplet is positioned between pixel 2 and pixel 3. At t = 0.005 s, the system transitions to the driving mode with both inner and outer electrodes applying a high voltage, resulting in the droplet’s movement to pixel 2. At t = 0.02 s, the system transitions to the detecting mode, where the inner electrode turns to a low level while the outer electrode maintains a high level. Following the cessation of voltage application to the inner electrode, the volume fraction of the droplet remains virtually unchanged, demonstrating its stable and sustained presence on the electrode surface. Simulation results confirm the robustness of the proposed electrode structure against perturbations and its effectiveness in driving and maintaining droplets. 

The parameters of circuit components and driving signals were verified in the simulation and are presented in Table 1. The droplet position sensing unit was constructed and simulated, employing the timing diagram shown in Figure 5a. The circuit functions in both driving and detection operational modes. Before transitioning into either operational mode, the circuit initiates a reset process by discharging the output terminals and sensing capacitors using the RST signal. In the driving mode, Gate is set to 55 V, and Data at 50 V, with respective pulse widths of 80 μs and 100 μs, ensuring effectual pixel voltage output. In the detection mode, Ctrl serves as the parallel column scanning signal, while Scan functions as the serial row scanning signal, enabling independent addressing and sensing of individual pixels. The timing sequence of RWS aligns with Scan, and RWS, Scan, and Ctrl each maintain a high voltage level of 50 V. The entire process is completed within 400 μs to detect droplets on individual pixels, thereby substantially enhancing system efficiency and demonstrating its potential for applications in large-scale arrays. 

Figure 5b illustrates the driving output of the proposed circuit. Within a span of 80 μs, C_po_ and C_pi_ can be charged to 48 V, highlighting exceptional charge–discharge capabilities. Furthermore, the circuit demonstrates a minimal voltage drop of less than 2 V after 1 ms, with the voltage remaining above 40 V for several tens of ms; this underscores its stable voltage retention capabilities, which are sufficient to meet the demands of driving and holding droplets in AM-EWOD systems. The capability of droplet position sensing of the proposed circuit has also been validated through simulation. Deionized water was utilized as the liquid sample, with an equivalent capacitance of approximately 0.12 pF, while silicone oil served as the surrounding medium with an equivalent capacitance of around 0.0015 pF. Meanwhile, to verify the circuit’s compatibility with other droplet samples, experiments were conducted using physiological saline (50 mg/mL, ε_r_ ≈ 67 [29]) and a phosphate-buffered saline (PBS) solution (ε_r_ ≈ 74 [30]), both of which are commonly used in biological experiments. The driving and detecting outputs of both samples are depicted in Figure 5c, where pi and po represent the voltages on the inner and outer electrodes, respectively. According to Equation (1), the output voltage exhibits a negative correlation with the magnitude of the equivalent capacitance; hence, the detection voltage for silicone oil is higher. The simulation results exhibit a substantial output variance exceeding 10 V, with silicone oil and deionized water registering 24.68 V and 11.13 V, respectively, validating the proposed circuit’s superior droplet position sensing capabilities. 

A 9 × 9 droplet array was constructed and simulated to explore the potential of the proposed unit for applications in large-scale array AM-EWOD. The driving voltage across the entire array is provided by row–column scanning signals Gate and Data. Concurrently, controlled by Scan and Ctrl, droplet voltages are detected in parallel outputs in each column. Figure 6a illustrates the driving voltages of the outer electrodes for pixels located at coordinates (0, 0) and (6, 8). It is clear that during the droplet detecting process, the voltage of po remains consistently around 48 V, demonstrating excellent driving stability. To clearly illustrate output variances between silicone oil and deionized water during array sensing, Column 8 was exclusively filled with either water or oil. The obtained array detecting outputs are also depicted in Figure 6a, presented from left to right, corresponding to the outputs from Row 0 to Row 1. It can be seen that even within a 9 × 9 array, the droplet sensing outputs exhibit a variance exceeding 10 V between deionized water and silicone oil; this demonstrates the potential of the proposed unit circuit for achieving array-based and miniaturized droplet position sensing in AM-EWOD systems.

A Shandong University (SDU)-shaped droplet was placed on the array for testing to validate the array’s on-site detection and independent addressing capabilities. Obtaining a graphical representation of droplet detecting results requires data processing of outputs, primarily because silicone oil exhibits higher detecting outputs, which can affect the observation of droplets. The measured detecting outputs were negated and then incremented by 20 V. These processed data are plotted according to pixel positions in Figure 6b, visually presenting the detected shapes of droplets. The detecting outputs of Column 0 are shown in Figure 6c, directly corresponding to Column 0 depicted in Figure 6b. 

In practical applications, continuous high voltage stress can induce a threshold voltage (V_th_) shift in a-Si TFTs, significantly impacting circuit stability, especially in precision-demanding applications such as droplet sensing. Therefore, threshold voltage shift stabilities of the proposed circuit were verified. According to Figure 3, after applying a positive bias for 3000 s, the fan-shaped TFT exhibited a V_th_ shift of merely 0.18 V. Additionally, during experiments, V_th_ can be recovered through heating. Consequently, a V_th_ range of −2 to 2 V was selected for simulation. Most TFTs in this circuit serve as switches, exhibiting a minimal Vth shift. Therefore, the analysis specifically targeted driving transistors T_2_ and T_8_. Figure 7a illustrates the pixel drive voltages and droplet sensing outputs as ΔV_th2_ and ΔV_th8_ vary from −2 V to 2 V. Since droplet driving does not pertain to silicone oil, the analysis focused solely on the pixel drive voltages for deionized water. Variations in ΔV_th2_ result in minor changes to the pixel drive voltages, with negligible effects on the droplet detection outputs because prior to droplet detection, signal RST discharges pi to a low level, thereby rendering the detecting voltage almost independent of the pixel voltage supplied by T_2_. Variations in ΔV_th8_ notably affect the droplet detection output, while the pixel driving voltage remains virtually unchanged because, in the driving mode, Scan and Ctrl signals are both maintained at a low level, effectively isolating T_8_ from the driving module. 

Figure 7b presents the statistical analysis of droplet sensing output under various ΔV_th8_ and the variation in pixel driving voltage under different ΔV_th2_. With the variation in ΔV_th2_, the change in the maximum pixel driving voltage is less than 0.5 V, while maintaining voltage as almost unchanged, thus demonstrating robust V_th_ shift stability. With the variation in ΔV_th8_, the oil and water output voltages exhibit linear changes, with their variance remaining almost constant because the detecting voltage equals the pi voltage after being charged by RWS, subtracted by V_th8_. This setup ensures that the difference in detecting voltages between oil and water depends solely on the former, which is influenced by the capacitance. Thus, while ΔV_th8_ may induce changes in the amplitude of detecting voltages, the consistent output variance between silicone oil and deionized water effectively mitigates the impact of threshold voltage, ensuring robust droplet sensing stability. 

In most on-chip biological experiments, it is essential to modify the environmental temperature, typically ranging from −20 to 60 °C. Temperature variations can significantly impact the performance of TFT, affecting the circuit’s output characteristics. Therefore, exploring the temperature stability of the proposed circuit is crucial. The characteristics of the a-Si:H TFT with a W/L ratio of 10 μm/6 μm were tested across various temperatures, and the transfer curve at V_ds_ = 41 V is shown in Figure 8a. As the temperature decreases, the threshold voltage of the TFT increases while both the on-state and off-state currents decrease. This degradation in performance can affect the circuit’s output characteristics. TFT models at different temperatures were extracted and employed to simulate the temperature stability of the circuit. Figure 8b illustrates the curves of driving outputs and detecting outputs at different temperatures. The driving voltage, detecting voltage, and variance between deionized water and silicone oil were extracted and plotted in Figure 8c. It is clear that with increasing temperature, both the driving voltage and the detecting voltages of oil and water exhibit a noticeable upward trend. Nevertheless, across the temperature range of −15 °C to 60 °C, the fluctuation in the driving voltage of the outer electrode does not exceed 2 V. Additionally, even at its minimum, the difference between the detecting voltages of oil and water remains greater than 10 V. This substantiates the circuit’s capability to maintain stable driving and droplet position sensing across varying temperatures.

## 4. Conclusions

This paper proposes an innovative capacitance-based pixel circuit integrating droplet driving and position sensing functionalities to mitigate the droplet movement crosstalk issue in large-scale AM-EWOD systems. This circuit features a unique dual-electrode pixel structure where discharge occurs exclusively on the inner electrode, effectively addressing the droplet vibration issues encountered in existing sensing circuits. The circuit employs a novel fan-shaped TFT structure. Threshold voltage drift and temperature stabilities of the circuit were validated. The circuit provides a stable driving voltage of around 48 V and demonstrates exceptional sensitivity by achieving a voltage variance exceeding 10 V when sensing deionized water and silicone oil. The proposed droplet sensing circuit offers an optimal solution for achieving closed-loop precise control of droplets in AM-EWOD systems. Nevertheless, challenges remain in streamlining the circuit structure and enhancing the speed of droplet sensing. Furthermore, the complexity of droplet sensing in real-world biological experimental settings adds additional challenges. Therefore, ongoing efforts are required to optimize circuit structures and enhance device performance. In future research, the proposed droplet position sensing circuit will demonstrate its effectiveness in on-chip experiments based on AM-EWOD. 

## Figures and Tables

**Figure 1 sensors-24-04789-f001:**
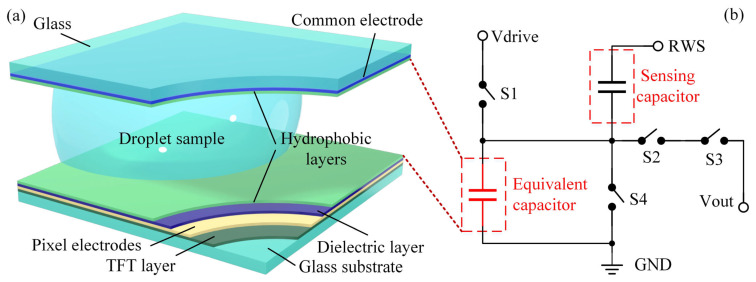
(**a**) Structure of TFT-based AM-EWOD device. (**b**) Simplified circuit model for droplet position sensing.

**Figure 2 sensors-24-04789-f002:**
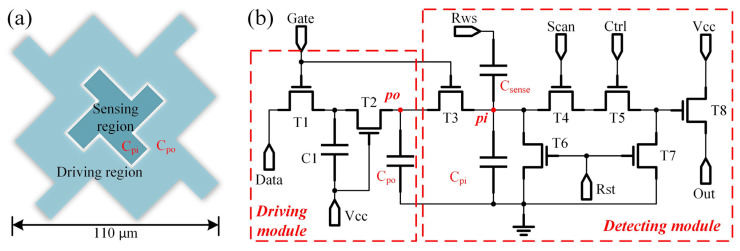
(**a**) Schematic diagram of novel dual-electrode pixel structure. (**b**) Diagram of droplet position sensing circuit.

**Figure 3 sensors-24-04789-f003:**
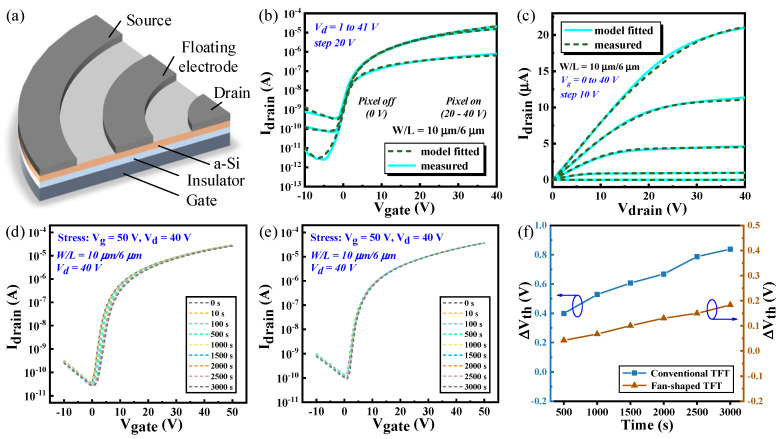
(**a**) Structure of fan-shaped a-Si:H TFT. (**b**) Transfer characteristics of standard a-Si:H TFT with W/L = 10 μm/6 μm. (**c**) Output characteristics of TFT. (**d**) Positive bias stability of conventional TFT. (**e**) Positive bias stability of fan-shaped TFT used in this work. (**f**) Threshold voltage shift with bias time.

**Figure 4 sensors-24-04789-f004:**
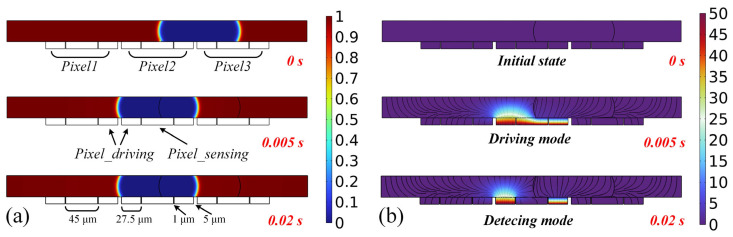
(**a**) Droplet volume fraction at different times. (**b**) Potential changes in droplet sensing system with dual-electrode structure in different operating modes.

**Figure 5 sensors-24-04789-f005:**
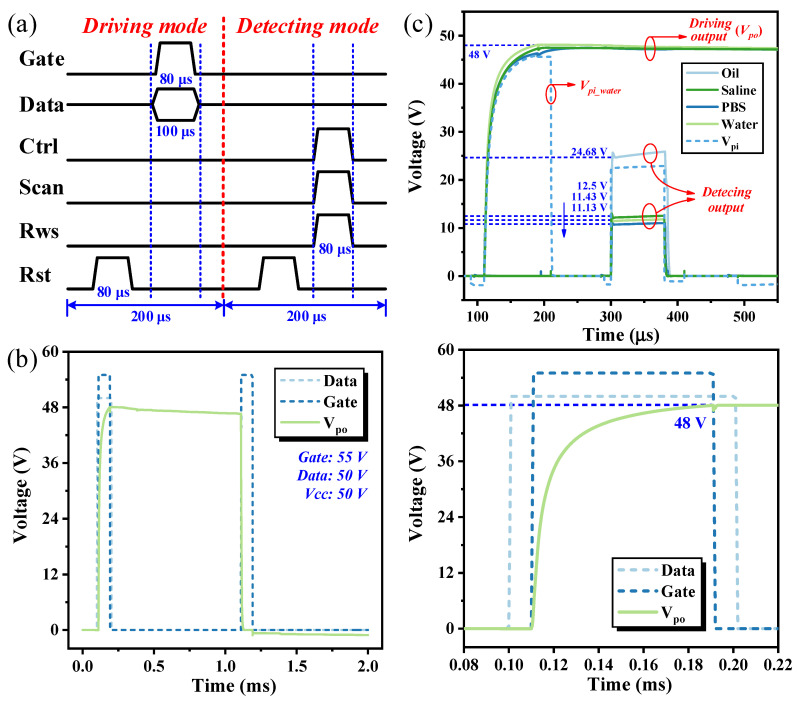
(**a**) Timing diagram of droplet position sensing unit. (**b**) Output of pixel driving mode. (**c**) Comparison of pixel driving and droplet detecting outputs for different liquid samples.

**Figure 6 sensors-24-04789-f006:**
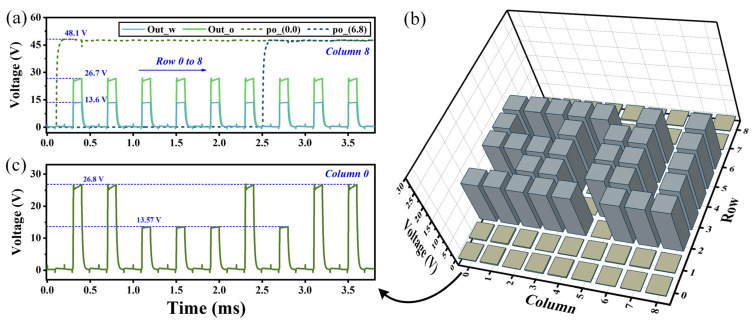
(**a**) Arrayed droplet detecting outputs for different liquid samples and driving voltages for water at different pixels. (**b**) Detecting outputs for SDU-shaped droplets on a 9 × 9 array. (**c**) Output of Column 0 in the 9 × 9 array.

**Figure 7 sensors-24-04789-f007:**
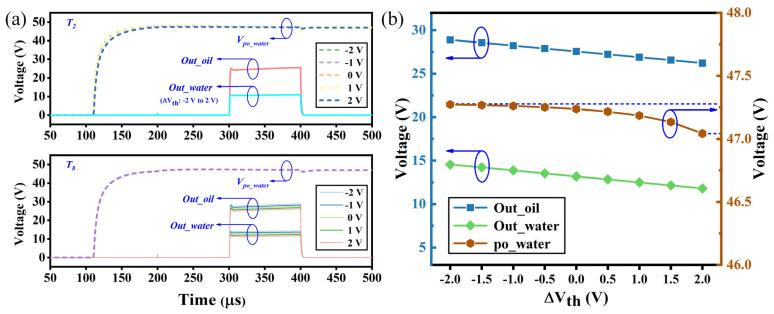
(**a**) Droplet sensing output and driving voltage on the outer electrode at different threshold voltages. (**b**) Influence of threshold voltage drift on the driving voltage and droplet sensing outputs.

**Figure 8 sensors-24-04789-f008:**
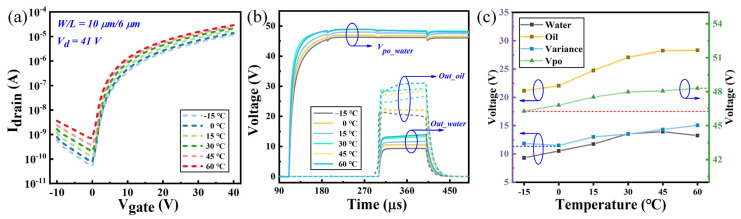
(**a**) Transfer characteristic curves of a-Si:H TFT at different temperatures. (**b**) Driving voltages and droplet sensing outputs at different temperatures. (**c**) Driving voltages, droplet sensing outputs, and output variance between oil and water at different temperatures.

**Table 1 sensors-24-04789-t001:** Parameters of the proposed droplet position sensing unit.

Device and Signal	Parameter
T_1_, T_3_, T_4_, T_5_, T_6_, T_7_	W/L = 10 μm/6 μm
T_2_, T_8_	W/L = 22 μm/6 μm
C_sense_	0.1 pF
C_1_	1 pF
Data, Vcc, Rws, Scan, Ctrl, Rst	50 V
Gate	55 V

## Data Availability

The study’s data are contained within the article.

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
