# Peer review of "Thin-Film Transistor Digital Microfluidics Circuit Design with Capacitance-Based Droplet Sensing"

_sensors, 2024, doi:10.3390/s24154789_

Round 1

Reviewer 1 Report

Comments and Suggestions for Authors

The paper proposes a novel capacitance droplet sensing system utilizing fan-shaped thin-film transistors (TFTs) with a dual-pixel electrode structure. This system achieves high pixel voltage and demonstrates good stability against threshold voltage drift and temperature fluctuations, with potential applications in large-scale arrays. Overall, it is a comprehensive paper covering system design, simulation, and stability evaluation. I have the following questions:

  1. The paper uses silicone oil and DI water as examples. Can the authors comment on the compatibility of this system with other common fluids?
  2. Can the authors elaborate on the benefits of the fan-shaped TFT design? A comparative analysis between the fan-shaped TFT and a conventional design would help illustrate the advantages in high-voltage performance and stability. Including specific metrics to evaluate stability would be beneficial. Currently, only the performance of the fan-shaped TFT is shown in Figure 3. Benchmarking against traditional back-gated transistors and drain-offset TFTs in terms of pixel voltage, self-heating effects, and stability would be useful.
  3. The system stability is evaluated based on a Vt shift range of -2 to 2V. Can the authors justify this chosen range of Vt shift? Is this the maximum range observed in Si TFTs?

Reviewer 2 Report

Comments and Suggestions for Authors

This paper presents a new capacitance-based circuit for droplet control that is applicable in digital microfluidics (DMF) based on active matrix electrowetting-on-dielectric (AM-EWOD) technology. The circuit achieves closed-loop control of individual droplets in a microfluidic channel, as it is able to drive the droplet movement and sense the position of the droplet in real time. It uses a dual-pixel electrode structure, with the discharge occurring only on the inner electrode during the droplet position sensing, thus mitigating droplet vibration issues and crosstalk during droplet movement. The circuit also uses a fan-shaped structure of thin-film transistors (TFTs) that enhances stability and performance, as it avoids self-heating effects through the more homogeneous distribution of the electric field.

In my opinion the paper fits within the scope of the journal but it is written in a way that is confusing and difficult to understand. I have some comments below but the main point is that the paper should be written much more clearly.

Comments:

·         The motivation section could be written better, with a reference to the literature to justify why the droplet vibration problem is important and needs addressing.

·         Lines 85-86: What are the dielectric and hydrophobic layers used? The materials are not mentioned.

·         Fig 1: The way this figure is structured and explained is confusing to me. There is no 1-to-1 correspondence between subfigures a and b. Hence, some of the things that exist in Fig. 1b (e.g., Vdrive, RWS) are not seen in Fig. 1a.

·         The sensing mechanisms are a bit confusing to me. Do we have two kinds of sensing going on (sensing of the type of liquid and position sensing), or just one (position sensing)? I am asking because when distinguishing between water and silicone oil, the capacitance of the droplet itself changes (due to the dielectric constant being different). Hence, this results in the “equivalent capacitor” changing because that capacitor includes the droplet (see Fig 1). But the paper mentions that the “sensing capacitor” is used for the position sensing of the droplet, and it is not clear what the “sensing capacitor” corresponds to in Fig. 1a. So, do we have two sensing mechanisms or one? In any case, I think this part of the paper should be written much more clearly.

·         Fig. 3: It would help to add additional subfigures, showing the distribution of the electric field lines in conventional TFT architectures vs the fan-shaped architecture, to illustrate the advantage.

·         Fig. 7a: The Vth changes are minor but over a scale of microseconds. What happens over longer timescales, with the device operating e.g., for 30 minutes?

·         I am not really sure if the simulation results add anything to the paper.
